# GC-MS-Identified Alkamides and Evaluation of the Anti-Inflammatory, Antibacterial, and Antioxidant Activities of Wild *Acmella radicans*

**DOI:** 10.3390/ijms26167884

**Published:** 2025-08-15

**Authors:** Israel Hurtado-Díaz, Rubicela Teta-Talixtacta, Antonio Bernabé-Antonio, José Antonio Silva-Guzmán, María Crystal Columba-Palomares, Silvia Marquina-Bahena, Mariana Sánchez-Ramos, Francisco Cruz-Sosa

**Affiliations:** 1Department of Molecular Biology and Genomics, University Center for Health Sciences, University of Guadalajara, Sierra Mojada 950, Col. Independencia Oriente, Guadalajara 44340, Jalisco, Mexico; israel.hurtado@academicos.udg.mx; 2Department of Wood, Pulp and Paper, University Center of Exact Sciences and Engineering, University of Guadalajara, Km 15.5 Guadalajara-Nogales, Col. Las Agujas, Zapopan 45100, Jalisco, Mexico; rubicelateta@gmail.com (R.T.-T.); jantonio.silva@academicos.udg.mx (J.A.S.-G.); 3Faculty of Pharmacy, Autonomous University of the State of Morelos, Av. Universidad 1001, Col. Chamilpa, Cuernavaca 62209, Morelos, Mexico; cpmc_ff@uaem.mx; 4Chemical Research Center-IICBA, Autonomous University of the State of Morelos, Av. Universidad 1001, Chamilpa, Cuernavaca 62209, Morelos, Mexico; smarquina@uaem.mx; 5Department of Biotechnology, Autonomous Metropolitan University-Iztapalapa Campus, Av. Ferrocarril de San Rafael Atlixco 186, Col. Leyes de Reforma 1ª. Sección, Alcaldía Iztapalapa, Mexico City 09310, Mexico; marianasan_06@xanum.uam.mx

**Keywords:** wild plant, extracts, fractions, biological activities, alkamides

## Abstract

*Acmella radicans*, commonly known as the “toothache plant,” is traditionally attributed with medicinal properties, although few studies have validated its biological effects. In the present study, a chemical analysis of the wild plant was performed using gas chromatography–mass spectrometry (GC-MS). In addition, the antioxidant, anti-inflammatory, and antibacterial potential of ethanolic extracts from the roots (RE) and aerial parts (AE), as well as their respective fractions, was evaluated. The dichloromethane fractions of the aerial parts (DFAE) and root extracts (DFRE) at a concentration of 25 μg/mL demonstrated the highest inhibition of nitric oxide (NO) production, reducing levels to 22.2 ± 1.9 and 22.2 ± 2.9 μM, respectively. Moreover, these fractions exhibited a notable inhibition of TNF-α production, lowering its concentration to 22.6 ± 3.3 pg/mL (DFAE) and 24.8 ± 5.3 pg/mL (DFRE) at 25 µg/mL. GC-MS chemical profiling revealed the presence of alkamides such as *N*-isobutyl-2*E*,6*Z*,8*E*-decatrienamide, *N*-(2-methylbutyl)-2*E*,6*Z*,8*E*-decatrienamide, and *N*-(2-phenylethyl)-2*E*,4*Z*-octadienamide in both root and aerial part extracts. The dichloromethane fractions showed a higher abundance of alkamides compared to the hexane fractions, suggesting that these compounds may be at least partially responsible for the observed anti-inflammatory activity. Additionally, AE showed moderate activity against *S. typhimurium* and low activity against other bacteria, while RE was especially effective against a resistant strain of *S. aureus*, indicating an MIC of 31.25 μg/mL, likely due to its high content of alkamides, particularly spilanthol. Several fractions also inhibited bacteria such as *P. aeruginosa*, *S. aureus*, and *E. coli*, possibly because of the presence of alkamides and compounds like *β*-amyrin.

## 1. Introduction

Since the beginning of civilization, medicinal plants have played a transcendental role, constituting a fundamental component of biocultural heritage and serving as the principal therapeutic resource in the primary care of multiple diseases [1,2]. In recent decades, the use of these phytotherapeutic resources has experienced a notable increase, motivated by the search for therapeutic alternatives that present greater tolerability and lower toxicity compared to conventional drugs [3]. In this context, the Asteraceae family is characterized by its wide geographical distribution and high content of secondary metabolites, which has led to growing interest in the pharmaceutical and food industries [4]. The genus *Acmella*, belonging to this family, is commonly named the “toothache plant” due to its ability to alleviate dental pain by inducing gingival paresthesia. Traditionally, various species of the genus have been used in dental pain therapy, with an emphasis on the use of roots and, to a lesser extent, leaves [5]. Moreover, species such as *Spilanthes acmella* and *Acmella oleracea* are used in infusions for the treatment of dysentery and in therapeutic baths for the relief of rheumatism. The flowers are also used in traditional medicine as a remedy for childhood stuttering and jaw swelling, while the seeds are considered stimulants and are used to treat common conditions such as colds, fever, and coughs [6,7].

Phytochemical studies have reported that the genus *Acmella* constitutes a significant source of bioactive compounds such as flavonoids, flavones, isoflavones, anthocyanins, catechins, polyphenols, and alkylamides. These compounds have been associated with multiple pharmacological activities, including antibacterial, anti-inflammatory, wound healing, antipyretic, antihemorrhagic, hepatoprotective, antispasmodic, antitumoral, antiparasitic, antifungal, and antioxidant effects [8,9,10]. The predominant compounds identified in the hexane extract of *Acmella uliginosa* were *N*-isobutyl-2*E*,6*Z*,8*E*-decatrienamide (37.80%), α-pinene (4.98%), and methyl hexadecanoate (4.78%) [11]. Furthermore, aliphatic alkylamides have been reported in the chloroform fraction of *Acmella ciliata* flowers [12]. The methanolic extract of *A. uliginosa* demonstrated superior antioxidant capacity compared to ethyl acetate and hexane extracts [11]. In *A. oleracea*, anti-inflammatory properties are primarily attributed to the alkylamide spilanthol; additionally, it has been confirmed that extracts from the leaves and flowers of *A. oleracea* contain alkylamides exhibiting anti-inflammatory activity in both in vitro and in vivo experimental models [13].

In a study, it was shown that extracts containing spilanthol, as well as isolated spilanthol from *A. oleracea*, can modulate the expression of inflammatory enzymes and mediators. Moreover, these extracts attenuate the activity of chymase, nitric oxide (NO), catalase (CAT), and superoxide dismutase (SOD), as well as chymase expression in vascular smooth muscle cells (VSMCs). Additionally, these significantly inhibit edema formation, NO production, and cellular infiltration in inflamed tissues in the formalin test in rats [14]. In another in vitro study using RAW 264.7 cells, spilanthol was shown to reduce the mRNA and protein expression levels of inducible nitric oxide synthase (iNOS) and cyclooxygenase-2 (COX-2) induced by lipopolysaccharide (LPS), suggesting that this compound inhibits the production of pro-inflammatory mediators at both the transcriptional and translational levels. Furthermore, it decreases the production of pro-inflammatory cytokines such as IL-1β, IL-6, and TNF-α. LPS-induced phosphorylation of the cytoplasmic inhibitor of NF-κB (IκB) and the nuclear DNA-binding activity of the transcription factor NF-κB are also reduced, partly due to NF-κB inactivation, which leads to the downregulation of pro-inflammatory mediator expression [15]. In another recent in silico and in vitro study of *A. paniculata,* it demonstrated anti-inflammatory potential, in which the hydroethanolic extract of its flowers (HEFeAP) demonstrated significant anti-inflammatory activity, as evidenced by its ability to inhibit albumin denaturation and to reduce levels of inflammatory mediators such as tumor necrosis factor-alpha (TNF-α) and NO in LPS-stimulated PBMC-derived macrophages. Molecular docking simulations revealed that the active phytochemicals in HEFeAP including coumarins, flavonoids, polyphenols, steroids, and anesthetic alkylamides bind strongly and stably to inflammatory biomarkers [16].

On the other hand, complete inhibition of the growth of *Enterobacter aerogenes* was observed using chloroform, petroleum ether, and methanol extracts of *Acmella paniculata* [17]. Similarly, the chloroform fraction of *A. ciliata* extracts inhibited the growth of *Escherichia coli*, *Staphylococcus aureus*, *Staphylococcus epidermidis*, and *Candida albicans*, an effect attributed to the presence of bioactive amides [12]. Ethanolic extracts from *A. oleracea* flowers, as well as their subsequent fractions (hexane, chloroform, ethyl acetate, and methanol), were subjected to phytochemical analysis, which confirmed the presence of spilanthol via thin-layer chromatography using Dragendorff’s reagent. These extracts exhibited antimicrobial activity, whereas the methanolic fraction additionally demonstrated significant antioxidant capacity [18,19]. On the other hand, limited studies have identified spilanthol as the principal alkylamide, particularly concentrated in the roots, with this molecule being attributed to anesthetic effects that alleviate dental, gingival, and pharyngeal pain [20,21].

In the absence of scientific reports on this subject, in the present study, a phytochemical profile of extracts and fractions of the wild *A. radicans* plant was performed and the anti-inflammatory, antibacterial, and antioxidant activities were evaluated.

## 2. Results and Discussion

### 2.1. Yield of Extracts and Fractions

The crude ethanolic extract of the aerial parts (AE) yielded 214.81 g (6.13%) of a semi-liquid, dark green viscous extract. In comparison, the crude ethanolic extract of the roots (RE) yielded 5.52 g (5.23%) of an extract with similar appearance and consistency to those of AE. The liquid–liquid extraction of AE or RE with various solvents resulted in a 40.20% dry weight yield for the aqueous fraction (AFAE), 34.80% for the dichloromethane fraction (DFAE), and 25.00% for the hexane fraction (HFAE). By contrast, the partitioning of RE resulted in yields of 9.38% for the hexane fraction (HFRE), 16.63% for the dichloromethane fraction (DFRE), and 73.97% for the aqueous fraction (AFRE).

### 2.2. Chemical Profile of Extracts and Fractions

Gas chromatography–mass spectrometry (GC-MS) analysis revealed distinct chemical profiles in the AE (compounds **1** to **18**) and for RE (compounds **1**, **3**, **5**, **6**, **10**, **12**, **15**, and **16**) (Table 1). Among the most characteristic components, alkamides were identified with notable relative abundances. For example, in AE, 3.65% of *N*-isobutyl-2*E*,6*Z*,8*E*-decatrienamide or spilanthol (**3**), 0.43% of *N*-(2-methylbutyl)-2*E*,6*Z*,8*E*-decatrienamide (**6**), 9.05% of *N*-(2-phenylethyl)-2*E*,4*Z*-octadienamide (**12**), and 7.78% of 3-phenyl-*N*-(2-phenylethyl)-2-propenamide (**14**) were found (Figure 1, Table 1).

In RE, compound **14** was not detected, whereas a high abundance of compound **3** was observed, accounting for 34.39% of the total composition. Compound **1** was detected in both extracts, consistent with the findings reported [20]. Additionally, the triterpenes *β*-amyrin (**17**) and lupeol acetate (**18**) were identified exclusively in AE (Table 1). The AE extract also contained ethyl and methyl esters of palmitic, linoleic, and linolenic acids.

Alkamide **3** was detected in all analyzed fractions, with the highest concentrations observed in DFAE (6.65%) and DFRE (46.34%), while lower levels were found in the hexane fractions. Alkamide **6** was also present in most fractions, particularly abundant in DFRE (9.98%), but was not detected in HFAE. Interestingly, HFAE contained nearly all the compounds identified in the AE extract, except for alkamides **6** and **12**. Alkamide **12** was exclusively detected in DFAE, where it accounted for 16.84% of the total composition. HFAE was distinctive in retaining phytol (**9**), stigmasterol (**15**), *γ*-sitosterol (**16**), and the triterpene lupeol acetate (**18**) as predominant constituents—compounds that were absent from the other fractions (Table 1).

### 2.3. In Vitro Anti-Inflammatory Activity

#### 2.3.1. Cell Viability

The cytotoxic potential of ethanolic extracts from the aerial (AE) and root (RE) parts of *A. radicans*, as well as their respective fractions of hexane (HFAE and HFRE), dichloromethane (DFAE and DFRE), and aqueous fractions (AFAE and AFRE), was assessed using LPS-stimulated RAW 264.7 macrophages. Treatments were applied at concentrations of 12.5, 25, 50, and 100 µg/mL, and cell viability was determined (Figure 2A–D and Figure 3A–D).

The AE ethanolic extract exhibited no significant cytotoxicity at concentrations up to 50 µg/mL, maintaining cell viability above 80% (Figure 2A). However, at 100 µg/mL, a marked reduction in viability was observed, with statistically significant differences (**** *p* < 0.0001), suggesting dose-dependent cytotoxicity effects likely due to bioactive constituents present at higher concentrations. By contrast, the RE ethanolic extract exhibited no significant cytotoxicity even at 100 µg/mL (Figure 3A).

The hexane (HFAE and HFRE) and aqueous (AFAE and AFRE) fractions exhibited no cytotoxicity at any of the concentrations, with cell viability consistently exceeding 90% (Figure 2B,D and Figure 3B,D), and even hexane fractions (12.5–50 μg/mL) exceeding 100% viability. These findings suggest that the more lipophilic compounds present in both aerial and root tissues are well tolerated by macrophages and may even exert a mild cytoprotective effect.

On the other hand, DFAE significantly decreased viability to 14.9 ± 3.1% at the highest concentration (Figure 2C). By contrast, lower concentrations of DFAE (≤25 μg/mL) did not show statistically significant differences compared to the LPS-stimulated control, indicating limited cytotoxicity at these doses. Similarly, DFRE exhibited a moderate reduction in cell viability at 50 μg/mL (79.8 ± 5.3%), while a pronounced cytotoxic effect was observed at 100 μg/mL, reducing viability to 14.9 ± 3.0% (Figure 3C).

Overall, these results indicate that the hexane and aqueous fractions from both plant parts AE and RE do not exhibit cytotoxicity and are suitable for subsequent anti-inflammatory and antimicrobial studies. Conversely, the AE and dichloromethane fractions should be used with caution at concentrations above 50 µg/mL to avoid compromising cell viability. These findings are essential for defining safe and effective experimental parameters for the biological evaluation of *A. radicans* extracts and their fractions. In another study, it was demonstrated that exposure to hexane and chloroform extracts of *Spilanthes acmella* at a concentration of 80 μg/mL for 24 h resulted in a marked reduction in cell viability, reaching 75 and 81%, respectively [15]. Conversely, the ethyl acetate and butanol extracts did not induce significant cytotoxicity, as evidenced by cell viabilities of 91 and 93%, respectively. Similarly, Stein et al. [14] reported that ethanolic extracts (25–100 μg/mL) derived from the leaves and flowers of *Acmella oleracea*, as well as isolated spilanthol (25–200 μM), did not exert significant cytotoxic effects on vascular smooth muscle cells (VSMCs), with viability levels comparable to those observed in the vehicle control (0.5% DMSO).

#### 2.3.2. Inhibition of NO Production

The inhibition of NO production was dependent on the chemical composition of the extract or fraction, as well as on the concentration used, as shown in Figure 4A–C and Figure 5A–C.

The NO concentration in the culture medium of LPS-stimulated cells was 43.5 ± 3.5 µM, significantly higher than that of control cells cultured with medium alone (8.3 ± 1.3 µM), indicating an almost fivefold increase in NO production upon LPS stimulation. Treatment with the AE extract resulted in concentration-dependent inhibition of NO production, with the highest inhibition observed at 50 µg/mL (12.2 ± 1.9 µM) (Figure 4A). Among the AE fractions, DFAE exhibited the most potent inhibitory effect, reducing NO levels to 22.2 ± 1.9 µM at a concentration of 25 µg/mL (Figure 4C). Notably, DFAE at 12.5 µg/mL also significantly inhibited NO production (31.0 ± 3.0 µM), a value comparable to that obtained with indomethacin (30.9 ± 2.3 µM). HFAP also demonstrated marked NO inhibition, reaching 21.8 ± 3.2 and 13.4 ± 1.4 µM at 50 and 100 µg/mL, respectively (Figure 4B). By contrast, AFAE showed minimal inhibitory activity, with a slight reduction in NO production only at the highest concentration tested of 100 µg/mL (Figure 4D).

On the other hand, ethanolic root extract (RE) exhibited a significant inhibition of NO production only at concentrations of 50 and 100 µg/mL, yielding nitrite levels of 32.0 ± 0.6 and 16.0 ± 3.6 µM, respectively (Figure 5A). The RE-derived fractions showed inhibitory activity comparable to that of the AE fractions. Specifically, at 25 µg/mL, both HFRE and DFRE significantly reduced nitrite levels to 36.3 ± 1.8 and 22.2 ± 2.9 µM, respectively (Figure 5B,C). AFRE also displayed inhibitory activity, although only at the highest concentration tested (100 µg/mL), resulting in a nitrite level of 37.2 ± 1.2 µM (Figure 5D).

To date, the anti-inflammatory potential of *A. radicans* has not been examined. In *Spilanthes acmella*, however, the hexane and chloroform extracts (80 µg mL^−1^) also reduced NO production to 28 and 15%, respectively, whereas the ethyl-acetate and butanol extracts induced only moderate inhibition (64 and 77%) [15]. Because cell viability was not evaluated, it remains unclear whether these reductions reflected true pharmacological activity or extract-induced cytotoxicity. On the other hand, it showed that the ethanolic leaf and flower extracts of *A. oleracea* (50–100 µg mL^−1^) markedly suppressed high-glucose-induced NO production in vascular smooth muscle cells [14]. Consistent with these in vitro findings, it was demonstrated that oral administration of an ethanolic *A. oleracea* extract (500 mg kg^−1^) significantly attenuated carrageenan-induced paw edema in Wistar albino rats during the acute, sub-acute, and chronic phases of inflammation [22].

#### 2.3.3. Inhibition of TNF-α Expression

The extracts and their respective fractions were also evaluated for their effects on TNF-α protein secretion, as determined by ELISA, in LPS-stimulated RAW 264.7 macrophages (Figure 6A–D and Figure 7A–D).

Extracts (AE and RE) and fractions (HFAE, AFAE, HFRE, and AFRE) were evaluated at concentrations of 25 and 50 µg/mL. By contrast, dichloromethane fractions (DFAE and DFRE) were tested at concentrations of 12.5 and 25 µg/mL, based on the cytotoxic effects observed at higher concentrations, as previously established in cell viability assays. Stimulation with LPS induced a marked increase in TNF-*α* production (45.2 ± 14.8 pg/mL) compared to untreated control cells (−1.9 ± 0.5 pg/mL). Treatment with 50 µg/mL of AE, HFAE, and HFRE resulted in a significant reduction in TNF-α levels to 18.8 ± 1.6, 22.2 ± 3.6, and 23.3 ± 2.1 pg/mL, respectively. These reductions were more pronounced than those achieved with the reference anti-inflammatory drug indomethacin (34.0 ± 8.1 pg/mL) (Figure 6A,B and Figure 7B). Moreover, the dichloromethane fractions DFAE and DFRE significantly reduced TNF-*α* production in a statistically significant manner at the concentration of 25 µg/mL (Figure 6C and Figure 7C).

This study constitutes the first report on the in vitro anti-inflammatory activity of both aerial parts (Figure 6) and roots (Figure 7) of the wild plant *A. radicans*. Within the genus *Acmella*, previous research has mainly focused on the Brazilian native species *A. oleracea* (syn. *Spilanthes acmella* var. *oleracea* or *Spilanthes oleracea*) [23].

In a previous study, methanolic extracts from the leaves and stems of *Spilanthes acmella* were reported to be non-cytotoxic in RAW 264.7 macrophages at concentrations up to 300 µg/mL and significantly inhibited NO production in LPS-stimulated cells. At this concentration, the extract also suppressed the production of PGE_2_ and reduced the expression of COX-2, as well as mRNA levels of pro-inflammatory cytokines, including IL-6 and IL-1β. By contrast, no significant inhibition of TNF-α production was observed. These anti-inflammatory effects were associated with the downregulation of mitogen-activated protein kinase (MAPK) pathways and nuclear factor kappa B (NF-κB) signaling [24].

The results reported by Wu et al. [15] align with our findings for *A. radicans*, in which the dichloromethane fractions of both aerial and root extracts exhibited the most pronounced inhibitory activity on NO production, followed by the hexane fractions, while the aqueous fractions showed only slight inhibition. Notably, TNF-α inhibition with hexane and dichloromethane fractions has been observed, suggesting that these non-polar extracts may contain key bioactive constituents responsible for modulating inflammatory responses [24].

In our study, alkamide-type compounds were predominantly identified in the dichloromethane fractions, which also exhibited the highest inhibitory activity on NO production. Among the alkamides detected, spilanthol was the most abundant, representing 34.39% of the total composition in the root extract (RE), and present in lower amounts in the aerial part extract (AE) (Table 1). Spilanthol has been widely reported for its anti-inflammatory properties. In *Spilanthes acmella*, this compound was shown to significantly reduce NO production and downregulate iNOS and COX-2 protein expression in LPS-stimulated RAW 264.7 macrophages, primarily through the inhibition of NF-κB activation, a key transcription factor involved in the regulation of pro-inflammatory mediators [15]. Supporting this, Freitas et al. [13] demonstrated that spilanthol also reduced the release of IL-8 and TNF-α in an LPS-activated neutrophil model.

In addition, the triterpenes *β*-amyrin and lupeol acetate were identified in AE and HFAE (Table 1). *β*-Amyrin has been reported to exert anti-inflammatory effects in microglial cells by significantly reducing LPS/IFN-γ-induced production and the expression of key pro-inflammatory markers, including TNF-α, IL-1β, IL-6, PGE_2_, and COX-2, as well as by decreasing NO levels [25,26]. Similarly, lupeol acetate has been shown to inhibit NO production in LPS-stimulated RAW 264.7 macrophages, with an IC_50_ value of 4.1 µM [27]. considering the presence of spilanthol, *β*-amyrin, and lupeol acetate in the most active fractions provides a strong chemical rationale for the observed anti-inflammatory effects of *A. radicans*.

### 2.4. Antibacterial Activity

The ethanolic extracts and their respective fractions exhibited distinct minimum inhibitory concentration (MIC) profiles when tested against different bacterial strains (Table 2).

The AE extract exhibited moderate activity exclusively against *S. typhimurium* (MIC = 62.5 μg/mL), while showing low activity (MIC = 125 μg/mL) against the other bacterial strains. By contrast, the RE extract demonstrated notable activity against *S. aureus* (R43300), a resistant strain (MIC = 31.30 μg/mL), suggesting potential effectiveness against antibiotic-resistant bacteria. GC-MS analysis revealed that the RE extract is predominantly composed of alkamide-type compounds, notably spilanthol (compound **3**), which may contribute significantly to the inhibition of *S. aureus* MRSA. The antimicrobial potential of alkamides is further supported by previous studies highlighting the efficacy of structurally related compounds such as capsaicin against *S. aureus* [28,29,30]. Among the evaluated fractions, HFAE, DFAE, DFRE, and AFRE exhibited inhibitory activity against *P. aeruginosa* and *S. aureus*, with MIC ≈ 62.5 μg/mL, while HFRE and DFRE also displayed comparable activity against *E. coli* (MIC ≈ 62.5 μg/mL). These antimicrobial effects may be associated with the higher content of alkamides observed in DFAE, HFRE, and DFRE. Additionally, HFRE also contained *β*-amyrin, a triterpenoid that may exert synergies or an additive antibacterial effect.

These findings align with previous studies. For example, Rincón-Mejía et al. [12] reported that an amide-rich chloroform fraction from *Acmella ciliata* inhibited *E. coli* growth using an MIC of 2500 μg/mL, while *S. aureus* and *S. epidermidis* were more susceptible, with an MIC of 1250 μg/mL. By contrast, aqueous extracts of *A. oleracea* have shown larger inhibition zones against various microorganisms in agar diffusion assays. Nevertheless, in the present study, aqueous fractions derived from ethanolic extracts did not inhibit bacterial growth when evaluated using the broth microdilution method. Similarly, in the case of *Acmella uliginosa*, the dichloromethane extract exhibited the highest antimicrobial activity against *S. aureus*, *S. aureus* (MRSA), *Staphylococcus epidermidis*, *E. faecalis*, and *P. aeruginosa*; however, the concentrations employed were relatively high (10 mg/mL), and the reported minimum inhibitory concentration (MIC) was 625 μg/mL [31]. For the ethanolic extracts of *A. oleracea*, various fractions were evaluated, among which only the chloroform fraction demonstrated inhibitory activity against *Salmonella typhi*, with an MIC of 31.25 μg/mL [18]. Finally, in another study, the diethyl ether extracts of *Acmella caulirhiza* showed the lowest MIC of 62.5 mg/mL against *Streptococcus mutans* (dental caries) [32].

When considered alongside the findings of the present study, these results suggest that alkamides may constitute some of the key bioactive constituents responsible for the antibacterial effects observed. Accordingly, *A. radicans* emerges as a valuable phytochemical source with significant potential for the discovery and development of novel antibacterial agents.

### 2.5. Total Phenolic and Flavonoid Content

Polyphenols constitute one of the principal classes of naturally occurring compounds in plants, characterized by the presence of at least one phenolic group in their chemical structure [33]. These compounds are associated with a wide range of health benefits, including protective effects against cardiovascular diseases, cancer, obesity, diabetes, and infectious diseases [34].

In the present study, the aerial part extract (AE) of *A. radicans* exhibited a total phenolic content (TPC) of 274 mg gallic acid equivalents (GAE)/100 g dry biomass (DB) and a total flavonoid content (TFC) of 38 mg quercetin equivalents (QE)/100 g DB. By contrast, the root extract (RE) showed a higher TPC of 363 mg GAE/100 g DB, but a markedly lower TFC of 5 mg QE/100 g DB.

In a related study, TPC values of 31.8 and 32.8 mg GAE/g extract were reported in ethanolic extracts from the roots and aerial parts of *A. radicans*, respectively. Additionally, TFC values of 113.7 mg QE/g extract in the roots and 317.5 mg QE/g extract in the aerial parts were observed [35]. In *A. oleracea*, the highest TPC was found in the leaves (7.59 mg GAE/g dry biomass), a value comparable to those obtained for *A. radicans* in the present study [36]. Similarly, a TFC of 72.14 mg QE/g extract and a TPC of 84.52 mg GAE/g extract were reported from the leaves of *Spilanthes acmella* [37]. A comparative phytochemical analysis showed that ethanolic leaf extracts of *Spilanthes ciliata* (syn. *Acmella ciliata*) exhibited a higher TPC (21.53 mg GAE/g dry biomass) than those of *S. oleracea* [38]. Furthermore, TPC quantification in various organs (leaf, root, stem, and flower) of *A. alba*, *A. oleracea*, and *A. calirrhiza* revealed lower TPC values in the roots relative to the leaves, particularly in *A. alba* and *A. oleracea* [39].

On the other hand, conventionally grown *A. oleracea* accumulated higher flavonoid contents in the leaves compared to hydroponically cultivated plants [40]. Moreover, in vitro cultures of *S. acmella* yielded significantly higher levels of both TPC (4.68 ± 0.70 mg GAE/g biomass) and TFC (4.03 ± 0.49 mg QE/g biomass) than those obtained from field-grown plants [41].

### 2.6. Antioxidant Activity

The ethanolic extract of the aerial parts of *A. radicans* exhibited a significantly higher antioxidant activity compared to the root extract. Specifically, antioxidant capacity values of 74 mM Trolox equivalents (TE)/100 g dry biomass (DB) and 84 mM TE/100 g DB were recorded using the DPPH and ABTS assays, respectively. By contrast, the antioxidant capacity of root extracts was approximately 50% lower in both assays, indicating a markedly reduced radical scavenging potential. In a related study, the antioxidant activity of *A. radicans* wild plant extracts was evaluated using the ABTS and DPPH methods [35]. Interestingly, their results indicated lower IC_50_ values for root extracts (142.9 µg/mL for DPPH and 26.2 µg/mL for ABTS), suggesting a higher antioxidant potency of the root compared to aerial parts under their specific conditions. These contrasting findings may reflect differences in extraction methods, environmental factors, or phenological stages of the plant material.

In other *Acmella* species, substantial antioxidant activity has also been reported. For instance, *A. oleracea* demonstrated high antioxidant potential in hydroalcoholic leaf extracts, with a DPPH value of 202.6 ± 5.8 mg TE/L [42]. The antioxidant activity of *A. alba*, *A. oleracea*, and *A. calirrhiza* was evaluated, and it was found that the flower and stem extracts exhibited the highest activity, with DPPH values ranging from 10.44 to 11.18 µmol TE/g DB and ABTS values from 11.97 to 16.74 µmol TE/g DB [39]. Similarly, flower extracts of *A. oleracea* showed strong antioxidant activity, with an IC_50_ of 89.6 µg/mL using the DPPH assay [43]. In *A. ciliata*, a strong relation was demonstrated between antioxidant activity and the levels of total phenolic and flavonoid compounds, with flower extracts showing the highest activity among the organs tested [44]. Despite these promising findings across the genus, *A. radicans* remains largely underexplored in terms of its phytochemical and biological properties. The results of the present study highlight the antioxidant potential of this species and underscore the need for further investigation to elucidate its bioactive composition and therapeutic relevance.

### 2.7. Limitations and Perspectives of the Current Study

One key limitation of this study is that all experiments were conducted exclusively in vitro, necessitating in vivo validation to assess the therapeutic efficacy, toxicity, pharmacokinetics, and safety of *A. radicans* extracts, fractions, and bioactive compounds. Although GC-MS analysis identified a high content of alkamides (e.g., spilanthol) and other bioactive molecules such as *β*-amyrin, these were not individually isolated or evaluated, limiting insight into their structure–activity relationships and mechanisms of action. Additionally, while reductions in inflammatory mediators like NO and TNF-α were observed, further research is needed to clarify the involved molecular pathways. The antibacterial assessment was limited to a few strains, indicating a need for broader spectrum testing, particularly against multidrug-resistant pathogens. Despite these limitations, the findings support the development of standardized formulations or enriched fractions with enhanced biological activity. Differences in responses between root and aerial part extracts suggest the potential for selective extraction strategies, paving the way for phytopharmaceutical development from *A. radicans*. Furthermore, cell, tissue, and organ cultures of this plant may be a sustainable future alternative to producing specific compounds of interest.

## 3. Materials and Methods

### 3.1. Plant Material

Whole plants (Figure 8) including roots, leaves, and flowers were collected in December 2021, in Tala, Jalisco, Mexico. Its geographic coordinates are 20°31′00″ north latitude and 103°41′23″ west longitude, at 1545 m above sea level. Taxonomic identification was determined by Dr. Daniel Sanchez Sánchez and Pablo Carrillo Reyes of the Laboratorio Nacional de Identificación y Caracterización Vegetal (LANIVEG-SECIHTI) and the “Luz Maria Villareal de Puga” Herbarium of the Instituto de Botanica of the Universidad de Guadalajara, México (IBUG). The specimen was confirmed as *Acmella radicans* (Jacq.) R.K. Jansen (Asteraceae) and deposited with registration No. 214556 (https://swbiodiversity.org/seinet/collections/individual/index.php?occid=30183465 (accessed on 12 August 2025).

### 3.2. Obtaining Crude Extracts

Roots and aerial parts were separated and then cut into pieces between 5 and 10 cm in length. Samples were dried in the shade at room temperature for one week. Dried material was pulverized in a blender, obtaining 3.5 kg of the aerial part and 105.40 g of the roots. Each biomass was macerated in 96% ethanol (Karal^®^; Karal, S.A. de C.V., Leon, Guanajuato, Mexico) in a 1:10 ratio (biomass/solvent), and four extractions were performed on the same sample for 72 h each. Extract solutions were filtered and concentrated in a BÜCHI EL 131 Rotavapor with a BÜCHI 461 Water Bath (BÜCHI Labortechnik AG, Flawil, Switzerland) to remove the solvent. Extracts were allowed to evaporate in a drying oven at 40 ± 2 °C to obtain a crude ethanolic extract of the aerial part (AE) and a crude ethanolic extract of the roots (RE).

### 3.3. Liquid–Liquid Extraction of the Crude Ethanolic Extracts

Crude extracts AE and RE were subjected to liquid–liquid extraction using hexane, dichloromethane, and water to obtain corresponding fractions. First, AE was resuspended with 300 mL of methanol in a 1 L separatory funnel and then 300 mL of hexane was added. The mixture was shaken vigorously and allowed to stand to form two phases, and the hexane phases were collected. The process was performed 3 times, and the hexane phase was concentrated to remove the solvent and obtain the hexane fraction (HF) from AE (HFAE). The methanolic phase was allowed to evaporate and was resuspended with 300 mL of water and then 300 mL of dichloromethane was added. The mixture was shaken and allowed to stand, and then the dichloromethane phase was collected, performing the process three times. The dichloromethane phase was concentrated to obtain the dichloromethane fraction (DFAE). The aqueous phase was evaporated in an oven at 40 °C and the aqueous fraction (AFAE) was obtained. For RE extract, the hexane (HFRE), dichloromethane (DFRE), and aqueous (AFRE) partitions were obtained in the same way using 100 mL of each solvent, and the yield for each fraction was obtained.

### 3.4. Analysis of Extracts and Fractions by GC-MS

The chemical profile of RE and AE and fractions (HFAE, DFAE, HFRE, and DFRE) was determined by gas chromatography coupled with mass spectrometry (GC-MS). An Agilent 6890 gas chromatograph with an Agilent 5973N (Agilent Technologies, Inc., Santa Clara, CA, USA) mass selective detector and an HP-5MS (5% phenyl)-methylpolysiloxane capillary column (30 m × 0.25 mm i.d., 0.25 μm film thickness) was used. The temperature ramp started at 100 °C for 1 min, and then was increased at 10 °C/min to 150 °C, then at 3 °C/min to 300 °C, and held for 4 min. Helium was used as the carrier gas at a flow rate of 1 mL/min. The injector temperature was 250 °C. An amount of 1 μL of the ethanolic fractions was injected at a concentration of 1 mg/mL and was injected in the “splitless” mode. Mass spectra were obtained using the electron impact method at 70 eV in a mass range of 20–600 DA. All peaks obtained were analyzed, and as many compounds as possible were identified by comparison with spectra from the NIST library version 1.7a. Peaks that could not be identified by the library were compared with spectra published in the literature. In addition, the relative abundance in percentage of each identified compound was determined by integrating the area under the curve of all peaks in the chromatograms.

### 3.5. In Vitro Anti-Inflammatory Activity of Crude Extracts and Fractions

#### 3.5.1. RAW 264.7 Macrophage Cell Culture

The RAW 264.7 murine macrophage cell line (Tib-71TM from ATCC^®^, Manassas, VA, USA) was cultured in DMEM/F12 ADVANCED (Gibco^TM^, Thermo Fisher Scientific Inc., Waltham, MA, USA) medium supplemented with 3.5% heat-inactivated fetal bovine serum (FBS) and 1% GlutaMax (Gibco^TM^, Thermo Fisher Scientific Inc., Waltham, MA, USA). Cells were maintained at 37 °C in a humidified 5% CO_2_ atmosphere and subcultured into 25 cm^2^ culture flasks. RAW 264.7 macrophages (20,000 cells/well in 0.2 mL of culture medium) were seeded into 96-well plates and incubated for 24 h under standard conditions. Cells were then pretreated with plant extracts or fractions at concentrations ranging from 12.5 to 100 µg/mL for 2 h. Dimethyl sulfoxide (DMSO; Sigma-Aldrich, St. Louis, MO, USA) at 0.45% (*v*/*v*) was used as the vehicle control and indomethacin (INDO; Sigma-Aldrich, St. Louis, MO, USA) at 7.5 µg/mL served as the positive control. Inflammatory stimulation was induced by the addition of lipopolysaccharide (LPS) at 1 µg/mL, followed by a 21 h incubation period. Experimental controls included a negative control (cells treated with culture medium only) and an LPS-stimulated control (without treatment). Following LPS exposure, 150 µL of cell-free supernatant was collected from each well for the quantification of nitric oxide (NO) and tumor necrosis factor-alpha (TNF-α) levels. The remaining culture medium was removed, and the cells were replenished with serum-free medium for immediate assessment of cell viability.

#### 3.5.2. Cell Viability Assay

Prior to the determination of nitric oxide (NO) production, cell viability was determined using 3-(4,5-dimethylthiazol-2-yl)-5-(3-carboxymethoxyphenyl)-2-(4-sulfophenyl)-2*H*-tetrazolium (MTS) reagent (Promega Corporation, Madison, WI, USA). An aliquot of 20 µL MTS reagent was added to each well containing cells and fresh medium without fetal bovine serum and incubated for 3 h. Subsequently, absorbance was measured at 490 nm in an ELISA plate reader. The average absorbance of the control cells stimulated only with LPS was considered as 100% cell viability.

#### 3.5.3. Determination of NO Concentration

Nitrite, a stable oxidation product of NO, was measured as an indirect indicator of NO production using the Griess reagent assay. Briefly, 50 µL of each supernatant was mixed with 100 µL of Griess reagent 50 µL of 1% sulfanilamide, and 50 µL of 0.1% *N*-(1-Naphthyl) ethylenediamine dihydrochloride in 2.5% phosphoric acid] for 10 min at room temperature. Subsequently, the absorbance at 540 nm was measured on an ELISA plate reader. Nitrite concentration was calculated using a calibration curve with NaNO_2_ prepared in fresh culture medium. The average absorbance of the reaction with the supernatants of the LPS-stimulated cell control (1 µg/mL) was considered as 100% of the NO production.

#### 3.5.4. Quantification of Cytokine TNF-α

TNF-α levels in the culture supernatants were quantified using a commercial ELISA kit (ELISA MAX™ Deluxe, BioLegend^®^, San Diego, CA, USA) according to the manufacturer’s protocol. The capture antibody was diluted in coating buffer and 100 μL was added to each well of a 96-well plate. Plates were sealed and incubated at 3–5 °C for 16–18 h to allow optimal antibody binding. Following incubation, plates were washed to remove unbound antibody, and nonspecific binding sites were blocked by adding 200 μL of blocking buffer per well. Plates were then incubated for 1 h at room temperature with shaking. After washing, 100 μL of either sample supernatants or serially diluted recombinant TNF-α standards was added to the wells to generate a standard curve. Plates were sealed and incubated with shaking at room temperature for 2 h. After washing, 100 μL of detection antibody solution was added to each well. Plates were sealed and incubated with shaking for 1 h at room temperature. Subsequently, plates were washed and 100 μL of avidin–horseradish peroxidase (HRP) conjugate was added to each well, followed by incubation with shaking for 30 min at room temperature. The wells were then washed, and 100 μL of freshly prepared 3,3′,5,5′-tetramethylbenzidine (TMB) substrate was added to initiate color development. Plates were incubated in the dark for 15 min, during which positive wells developed a blue coloration. The enzymatic reaction was terminated by adding 100 μL of 2N sulfuric acid, which caused a color shift from blue to yellow. Absorbance was measured at 450 nm using a microplate reader. All washing steps employed phosphate-buffered saline (PBS, pH 7.4) containing 0.05% Tween-20.

### 3.6. Antibacterial Activity of Crude Extracts and Fractions

All extracts and fractions were evaluated in an in vitro culture against *Escherichia coli* (ATCC 8739), *Streptococcus pyogenes* (ATCC 19615), *Pseudomonas aeruginosa* (isolated from clinical cases), *Staphylococcus aureus* (ATCC 6538), *Salmonella typhimurium* (ATCC 14028), and *Staphylococcus aureus* MRSA (ATCC 43300). The evaluation was performed by the microdilution method [45,46]. The extracts were used from 31.3 to 500 μg/mL dissolved in DMSO and then diluted in Mueller–Hinton broth (Difco, Detroit, MI, USA). Gentamicin (GEN) was used as a reference drug (Garamycin; Schering-Plough Corporation, Kenilworth, NJ, USA) from 0.15 to 40 µg/mL. Bacterial inoculum was prepared in 0.85% saline solution by adjusting the turbidity to 0.5 on the McFarland scale, corresponding to approximately 1.5 × 10^8^ CFU/mL. Subsequently, the suspensions were diluted to achieve a final concentration of 5 × 10^5^ CFU/mL. One row of wells was designated as the growth control, containing 100 μL of broth and 100 μL of bacterial inoculum. Another row was used as the sterility control, consisting of 200 μL of broth without inoculum. For the evaluation of potential antimicrobial agents, each test concentration was prepared by adding 100 μL of bacterial inoculum and 100 μL of the corresponding sample stock solution. DMSO at a final concentration of 10% (*v*/*v*) was used as the negative control. All experiments were performed in triplicate. The plates were covered and incubated at 37 °C for 24 h. Subsequently, absorbance readings were obtained using the Glomax multi-detection system (Promega Corporation, Madison, WI, USA) at 600 nm. The minimum concentration of antibacterial agent responsible for the inhibition of bacterial growth was defined as the minimum inhibitory concentration (MIC). After incubation, 10 μL of MTT (0.4 mg/mL) was added to each well and the samples were incubated for 3 h at room temperature to determine the bactericidal or bacteriostatic effect.

### 3.7. Determination of Total Polyphenolics and Antioxidant Activity of Crude Extracts

#### 3.7.1. Obtaining Extracts

Total polyphenolic content was determined by macerating 150 mg of ground dry biomass from the aerial parts or roots with 96% ethanol (Karal^®^) at a 1:20 ratio. Three extraction cycles were performed on each sample and were kept under constant agitation at 115 rpm for 72 h at 25 ± 2 °C. The extracts were concentrated in a BÜCHI EL–131 rotary evaporator (BÜCHI Labortechnik AG, Flawil, Switzerland), and the volume was made up to 20 mL with 96% ethanol. The extracts were used to determine the total phenolic content (TPC), total flavonoids content (TFC), and antioxidant activity.

#### 3.7.2. Determination of Total Phenolics and Flavonoids Content

TPC was determined by the Folin–Ciocalteu (FC) reagent (Sigma-Aldrich, St. Louis, MO, USA) method [47]. Briefly, 200 μL of extract was mixed with 1900 μL of distilled water and 200 μL of FC (2N). The mixture was shaken for 3 min and then 1000 μL of Na_2_CO_3_ (20%, *w*/*v*) was added. The reaction was incubated on an orbital shaker at 115 rpm for 1 h. Samples were immediately analyzed against a blank (containing all reagents without extract) at 765 nm using a Genesys 150 spectrophotometer (Thermo Fisher Scientific Inc., Waltham, MA, USA). Quantification was performed using a calibration curve of gallic acid (GA) from 25 to 400 mg/L. Results are expressed as mg GA equivalents (mg GAE) per g dry biomass (g DB) ± standard deviation. All determinations were performed in triplicate.

TFC was determined using the AlCl_3_ method [48]. Briefly, 200 μL of the ethanolic extract was mixed with 1250 μL of distilled water and 75 μL of NaNO_2_ (5% *w*/*v*). The mixture was incubated under shaking at 115 rpm for 6 min and then 150 μL of AlCl_3_ (10% *w*/*v*) was added. The reaction was incubated under shaking for 5 min and 500 μL of NaOH (1 M) was added. The solution was made up to 2.5 mL with distilled water. Samples were analyzed against a blank (containing all reagents without the extract) at 510 nm. Quantification was performed using a calibration curve of quercetin (QT) from 0.781 to 50 mg/L. Results are expressed as mg QT equivalents (mg QTE) per g dry biomass (DB) ± standard deviation.

#### 3.7.3. Determination of Antioxidant Activity by DPPH and ABTS

Antioxidant activity was determined by the 2,2-diphenyl-1-picrihydrazyl (DPPH) and 2,2′-azino-bis-3-ethylbenzothiazoline-6-sulfonic acid (ABTS) (Sigma-Aldrich, St. Louis, MO, USA) methods [49]. For DPPH, briefly, 1.9 mL of DPPH solution (ethanol or methanol) (0.1 mM) and 100 μL of extract were mixed; the reaction was incubated under shaking for 30 min and the absorbance at 517 nm was immediately measured. Quantification was performed using a calibration curve of Trolox (Tx) (Sigma-Aldrich, St. Louis, MO, USA) from 0.781 to 50 mg/L. The results are expressed in mg Tx equivalents (mg TxE) per g of dry biomass (DB) ± standard deviation; the experiment was performed in triplicate.

For ABTS, a stock solution was prepared by mixing a solution of ABTS (7 mM) with K_2_S_2_O_8_ at 2.45 mM in a 1:1 ratio; the mixture was kept in the dark under constant stirring at 115 rpm at 25 °C for 12–16 h. Subsequently, the absorbance was read at 734 nm and adjusted to 0.7 ± 0.2 using phosphate-buffered saline. Then, 1.9 mL of the above solution was mixed with 100 μL of extract and allowed to incubate under stirring for 6 min, and the absorbance was measured at 734 nM. Quantification was performed using a Trolox (Tx) calibration curve from 0.781 to 50 mg/L. Results are expressed as mg Tx equivalents (mg TxE) per g dry biomass (DB) ± standard deviation; the experiment was performed in triplicate.

### 3.8. Statistical Analysis

Data are presented as the mean ± standard deviation (SD). All experiments were performed 2–3 independent times, each including a minimum of three replicates per treatment. The data were tested for normality using the Shapiro–Wilk method and differences were considered statistically significant with a *p*-value < 0.05. Then, a statistical analysis was conducted using one-way ANOVA followed by Dunnett’s post hoc test (*p* < 0.05), implemented in GraphPad Prism version 8.0 prior to ANOVA.

## 4. Conclusions

This study highlights the pharmacological potential of *A. radicans*, emphasizing the bioactivity of its crude extracts (AE and RE) and dichloromethane fractions. These samples exhibited significant anti-inflammatory effects, demonstrated by decreased nitric oxide and TNF-*α* production in LPS-stimulated macrophages. The GC-MS analysis revealed a high content of alkamides, particularly spilanthol and its analogs, likely responsible for this activity. In addition, the root extract showed antibacterial activity against resistant *S. aureus*, while the HFRE, DFRE, and AFRE fractions were effective against *P. aeruginosa*, *S. aureus*, and *E. coli*, potentially due to the presence of alkamides and *β*-amyrin. The root extracts showed higher total phenolic content, while the aerial extracts demonstrated stronger antioxidant activity. The findings confirm that *A. radicans* possesses notable anti-inflammatory, antibacterial, and antioxidant activities, possibly attributed to its high alkamide content, particularly in the dichloromethane fractions of the root and aerial part extracts, supporting the traditional use of medicinal plants and highlighting its potential as a source of bioactive compounds. Further in vivo studies, the development of bioactive enriched formulations, and toxicity profiling have yet to be performed to validate and expand its therapeutic applications.

## Figures and Tables

**Figure 1 ijms-26-07884-f001:**
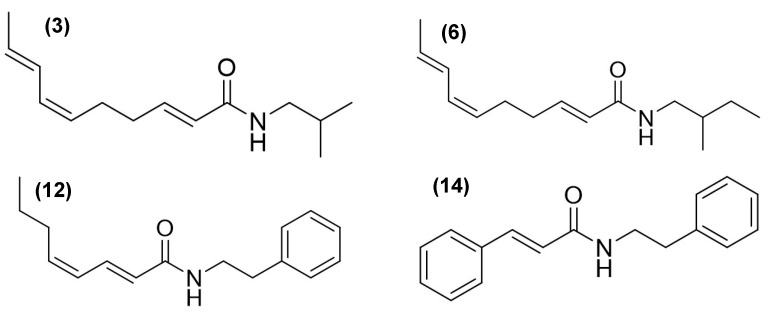
Alkamides identified by GC-MS in the ethanolic extracts of the aerial parts (compounds **3**, **6**, **12**, and **14**) and roots (compounds **3**, **6**, and **12**) of *A. radicans*.

**Figure 2 ijms-26-07884-f002:**
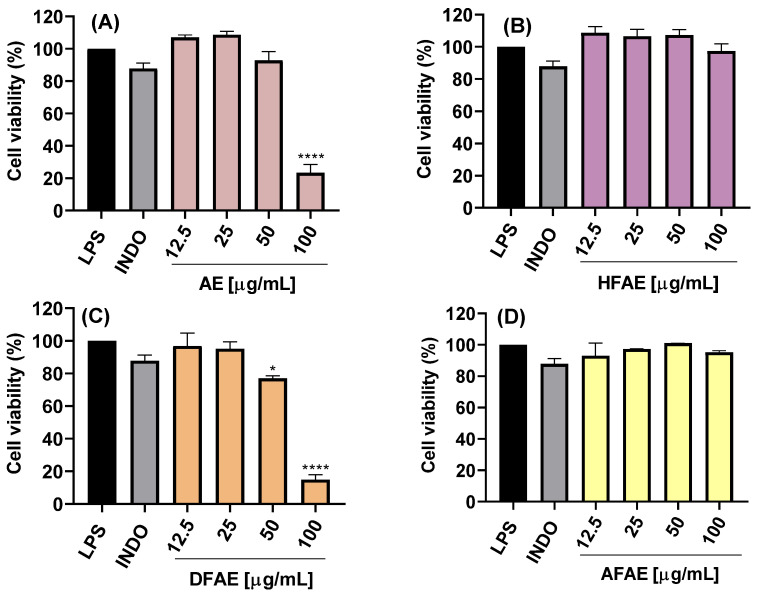
Effects of ethanolic extract and fractions from the *A. radicans* aerial part on RAW 264.7 cell viability. AE = ethanolic aerial part extract (**A**); HFAE = hexane fraction of AE (**B**); DFAE = dichloromethane fraction of AE (**C**); AFAE = aqueous fraction of AE (**D**). INDO = indomethacin (7.5 µg/mL); LPS = lipopolysaccharides (1 µg/mL). Data represents the mean ± standard deviation of three independent experiments, each in triplicate (n = 3). * *p* < 0.05, **** *p* < 0.0001 with respect to the control of cells treated only with LPS.

**Figure 3 ijms-26-07884-f003:**
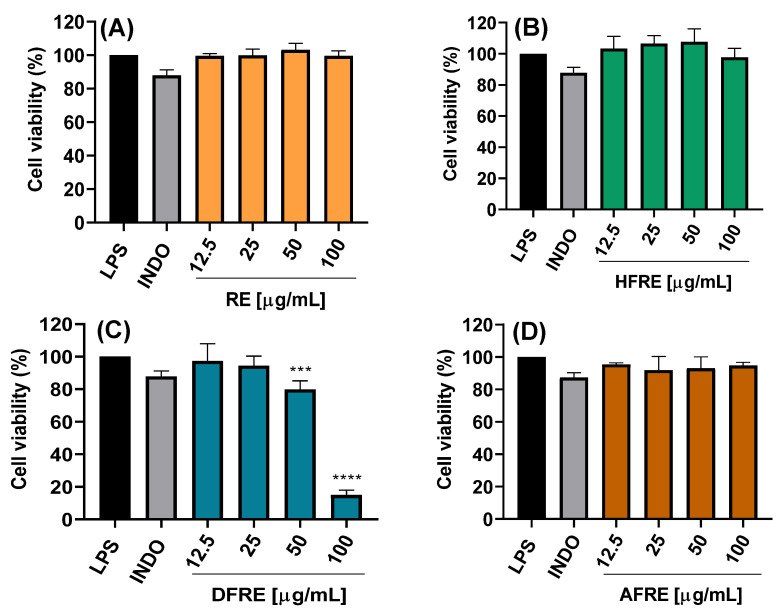
Effects of the extract and fractions from the *A. radicans* root on RAW 264.7 cell viability. RE = ethanolic root extract (**A**); HFRE = hexane fraction of RE (**B**); DFRE = dichloromethane fraction of RE (**C**); AFRE = aqueous fraction of RE (**D**). INDO = indomethacin (7.5 µg/mL); LPS = lipopolysaccharides (1 µg/mL). Data represents the mean ± standard deviation of three independent experiments, each in triplicate (n = 3). *** *p* < 0.001, **** *p* < 0.0001 with respect to the control of cells treated only with LPS.

**Figure 4 ijms-26-07884-f004:**
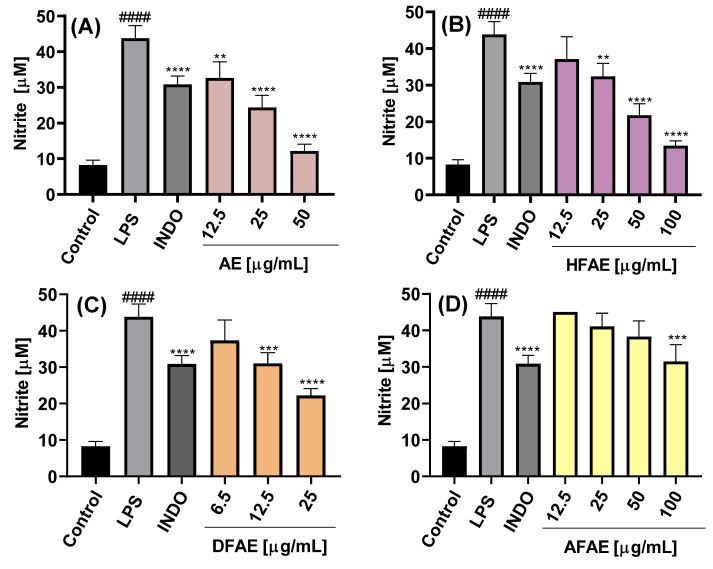
Effects of the extract and fractions from the *A. radicans* aerial part on the inhibition of NO production in RAW 264.7 cells. AE = ethanolic aerial part extract (**A**); HFAE = hexane fraction of AE (**B**); DFAE = dichloromethane fraction of AE (**C**); AFAE = aqueous fraction of AE (**D**). INDO = indomethacin (7.5 µg/mL); LPS = lipopolysaccharide (1 µg/mL). Data represents the mean ± standard deviation of three independent experiments, each in triplicate (n = 3). #### *p* < 0.0001 with respect to the LPS-free control cells. ** *p* < 0.01, *** *p* < 0.001, **** *p* < 0.0001 with respect to the control cells treated only with LPS.

**Figure 5 ijms-26-07884-f005:**
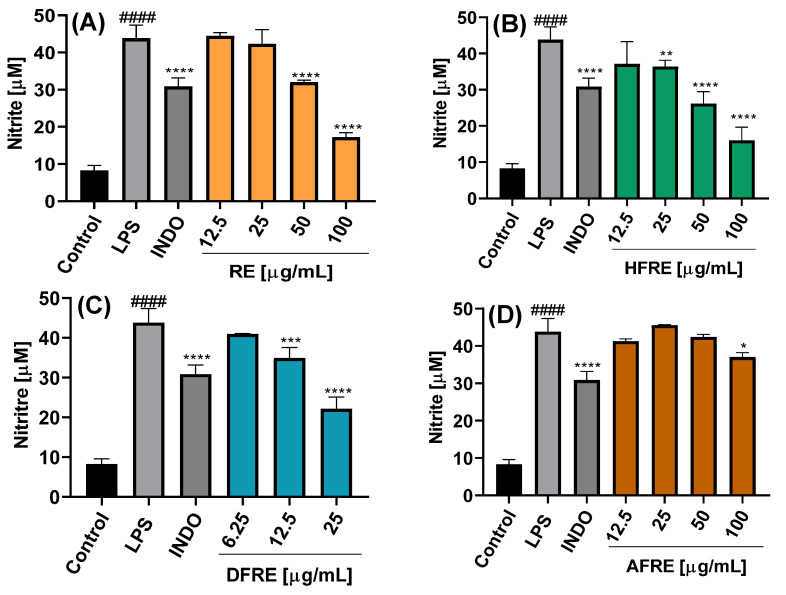
Effects of the extract and fractions from *A. radicans* roots on the inhibition of NO production in RAW 264.7 cells. RE = ethanolic root extract (**A**); HFRE = hexane fraction of RE (**B**); DFRE = dichloromethane fraction of RE (**C**); AFRE = aqueous fraction of RE (**D**). INDO = indomethacin (7.5 µg/mL); LPS = lipopolysaccharide (1 µg/mL). Data represents the mean ± standard deviation of three independent experiments each in triplicate (n = 3). #### *p* < 0.0001 with respect to the LPS-free control cells. * *p* < 0.05, ** *p* < 0.01, *** *p* < 0.001, **** *p* < 0.0001 with respect to the control cells treated only with LPS.

**Figure 6 ijms-26-07884-f006:**
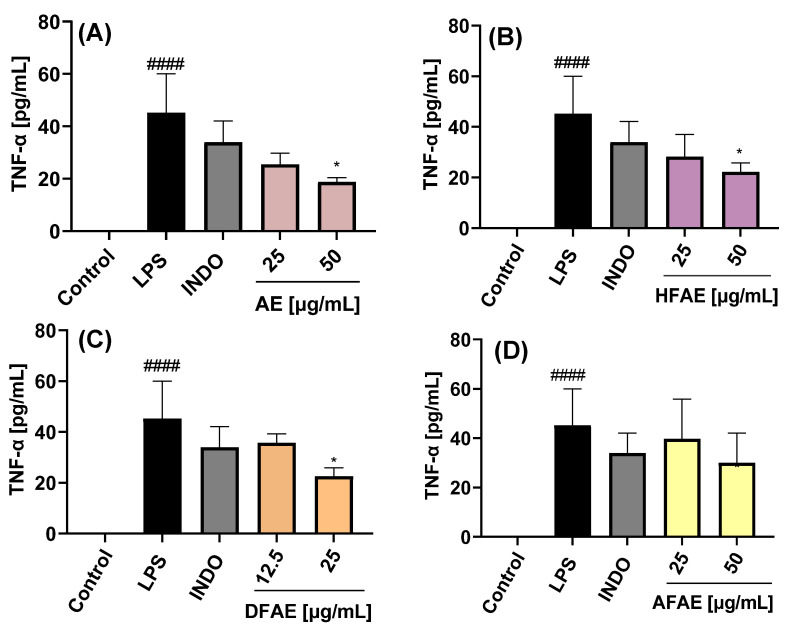
Effects of the aerial part extract and its fractions on the inhibition of TNF-α expression in LPS-stimulated RAW 264.7 cells. AE = ethanolic aerial part extract (**A**); HFAE = hexane fraction of AE (**B**); DFAE = dichloromethane fraction of AE (**C**); AFAE = aqueous fraction of AE (**D**). INDO = indomethacin (7.5 µg/mL); LPS = lipopolysaccharide (1 µg/mL). Data represents the mean ± standard deviation of three independent experiments, each in triplicate (n = 3). #### *p* < 0.0001 with respect to the LPS-free control cells. * *p* < 0.05 with respect to the control cells treated only with LPS.

**Figure 7 ijms-26-07884-f007:**
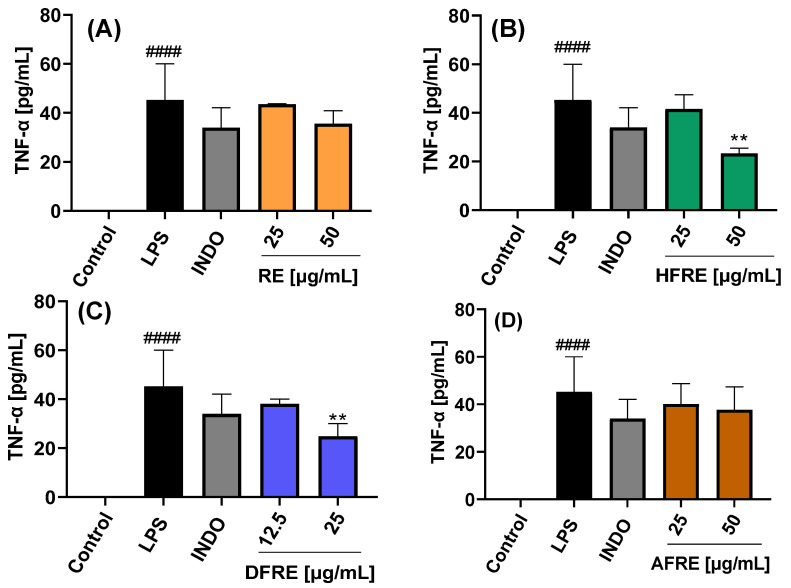
Effects of the root extract and its fractions on the inhibition of TNF-α expression in LPS-stimulated RAW 264.7 cells. RE = ethanolic root extract (**A**); HFRE = hexane fraction of RE (**B**); DFRE = dichloromethane fraction of RE (**C**); AFRE = aqueous fraction of RE (**D**). INDO = indomethacin (7.5 µg/mL); LPS = lipopolysaccharides (1 µg/mL). Data represents the mean ± standard deviation of three independent experiments each in triplicate (n = 3). #### *p* < 0.0001 with respect to the LPS-free control cells. ** *p* < 0.01 respect to the control cells treated only with LPS.

**Figure 8 ijms-26-07884-f008:**
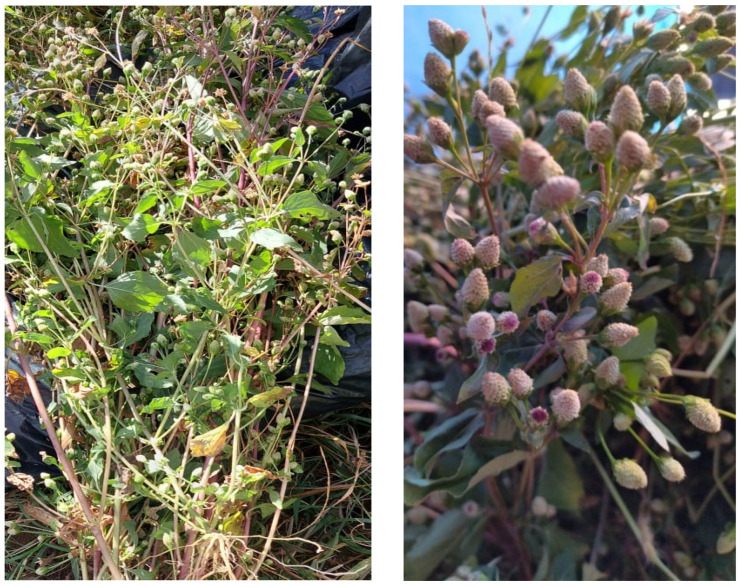
Adult specimens of *A. radicans* collected in the municipality of Tala, Jalisco, Mexico.

**Table 1 ijms-26-07884-t001:** The chemical profile of ethanolic extracts and fractions of the aerial part and roots of *A. radicans* analyzed by GC-MS.

No.	Compound Name	Chemical Formula	Retention Time (min)	Relative Abundance (%)					
				AE	RE	HFAE	DFAE	HFRE	DFRE
1	2-Tridecanone	C_13_H_26_O	13.77	1.79	1.28	4.29	n.d.	5.33	n.d.
2	Phytol acetate	C_22_H_42_O_2_	17.58	0.81	n.d.	0.61	n.d.	n.d.	n.d.
3	*N*-Isobutyl-2*E*,6*Z*,8*E*-decatrienamide (spilanthol)	C_14_H_23_NO	18.26	3.65	34.39	1.61	6.65	11.93	46.34
4	Palmitic acid methyl ester	C_17_H_34_O_2_	18.46	2.80	n.d.	0.82	n.d.	n.d.	n.d.
5	Palmitic acid ethyl ester	C_18_H_36_O_2_	19.13	4.22	3.43	6.47	n.d.	6.68	n.d.
6	*N*-(2-Methylbutyl)-2*E*,6*Z*,8*E*-decatrienamide	C_15_H_25_NO	19.32	0.43	6.22	n.d.	0.85	2.66	8.97
7	Linoleic acid methyl ester	C_19_H_34_O_2_	20.10	1.67	n.d.	2.60	1.71	n.d.	n.d.
8	Linolenic acid methyl ester	C_19_H_32_O_2_	20.16	1.17	n.d.	5.52	n.d.	3.88	n.d.
9	Phytol	C_20_H_40_O	20.28	2.73	n.d.	4.59	n.d.	n.d.	n.d.
10	Linoleic acid ethyl ester	C_20_H_36_O_2_	20.70	3.43	2.15	3.36	15.43	n.d.	9.62
11	Linolenic acid ethyl ester	C_20_H_34_O_2_	20.77	3.61	n.d.	2.62	n.d.	n.d.	n.d.
12	*N*-(2-phenylethyl)-2*E*,4*Z*-octadienamide	C_16_H_21_NO	21.34	9.05	4.85	n.d.	16.84	n.d.	n.d.
13	Pentacosane	C_25_H_52_	23.78	1.59	n.d.	2.37	n.d.	9.56	n.d.
14	3-phenyl-*N*-(2-phenylethyl)-2-propenamide	C_17_H_17_NO	23.91	7.78	n.d.	2.47	n.d.	5.49	n.d.
15	Stigmasterol	C_29_H_48_O	34.55	1.82	4.61	4.99	n.d.	n.d.	n.d.
16	*γ*-sitosterol	C_29_H_50_O	35.57	1.82	2.59	5.10	n.d.	n.d.	n.d.
17	*β*-Amyrin	C_30_H_50_O	38.23	2.99	n.d.	4.29	n.d.	5.33	n.d.
18	Lupeol acetate	C_32_H_52_O_2_	39.30	3.40	n.d.	0.61	n.d.	n.d.	n.d.

Ethanolic aerial part extract (AE); ethanolic root extract (RE); hexane fraction of AE (HFAE); dichloromethane fraction of AE (DFAE); hexanic fraction of RE (HFRE); dichloromethane fraction of RE (DFRE). n.d. = not detected.

**Table 2 ijms-26-07884-t002:** Antibacterial activity of ethanolic extracts and fractions of *Acmella radicans*.

Plant Source	MIC (μg/mL)					
	*Escherichia coli*	*Pseudomonas aeruginosa*	*Salmonella typhimurium*	*Staphylococcus aureus*	*Staphylococcus aureus*-MRSA	*Streptococcus pyogenes*
Extract						
AE	125	125	62.5	125	125	125
RE	125	125	125	125	31.3	125
Fraction						
HFAE	125	62.5	125	62.5	125	125
DFAE	125	62.5	125	125	125	500
AFAE	125	125	250	125	125	500
HFRE	62.5	125	125	125	500	125
DFRE	62.5	62.5	250	61.5	250	250
AFRE	125	62.5	125	61.25	500	500
Gentamicin *	4	2	4	1	≥16	4

AE = ethanolic aerial part extract; RE = ethanolic root extract; HFAE = hexane fraction of AE; HFRE = hexane fraction of RE; DFAE = dichloromethane fraction of AE; DFRE = dichloromethane fraction of RE; AFAE = aqueous fraction of AE, AFRE = aqueous fraction of RE. * antibiotic drug as a positive control. MRSA = Methicillin-resistant strain.

## Data Availability

Data are contained within the article.

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
