# Peer review of "GC-MS-Identified Alkamides and Evaluation of the Anti-Inflammatory, Antibacterial, and Antioxidant Activities of Wild Acmella radicans"

_ijms, 2025, doi:10.3390/ijms26167884_

Round 1
Reviewer 1 Report
Comments and Suggestions for Authors
Overall, the article is well-written, including methodogy, sufficient details for other scientist to perform the same. Authors have chosen Acmella radicans plant, with significant theraputic potential, given its use in traditional medicine. However, little evidence in the literature exist regarding its biological activies, both in vitro and vivo. So, the subject of the study is innovative, giving new info about a plant that can potentially be used as fitotherapeutic in the future. However, I have several requests for the authors, in order to improve the quality of the article:
- In the last paragraph of Introduction section, please single out only the main objectives of the study. The last two sentences look like conclusion , which can be confusing for readers. Please, revise this part. Other parts of Introduction are satisfactory, containing the most important findings regarding Acmella radicans regarding its components and biological activities. Authors have mentioned antioxidant, antibacterial and antiiflammatory action of this plant. Please, add some references regarding antiiflammatory action, including in vivo results.
- It is recommended to perform additional experiments regarding antioxidant potential, besides ABTS and DPPH. Eg. FRAP, inhibition of lipid peroxidation, O2-, H2O2, MDA….etc
- Please add some additional info about the normality of data distribution and tests used in the subsection: Statistical analysis
- In the conclusion section, please mention the main finding regarding phytochemical components and antioxidant potential, you have focused only on antiiflammatory and antibacterial effects of radicans
- The authors should discuss and compare the results of this study with others that not only investigated A.radicans but also its components alone. Try to explain which component is responsible for antioxidant, anti-inflammatory and antibacterial effect and the mechanisms of action as well
- Also, add future perspectives, and limitations of the study in the discussion section.
- Please, add n- values in the captions of figures
Can be improved, please adress a native speaker or a professional to go throught the whole manuscript to correct minor mistakes, grammar and spelling.
Author Response
Response to Reviewer 1 Comments
Overall, the article is well-written, including methodogy, sufficient details for other scientist to perform the same. Authors have chosen Acmella radicans plant, with significant theraputic potential, given its use in traditional medicine. However, little evidence in the literature exist regarding its biological activies, both in vitro and vivo. So, the subject of the study is innovative, giving new info about a plant that can potentially be used as fitotherapeutic in the future. However, I have several requests for the authors, in order to improve the quality of the article:
Comments 1. In the last paragraph of Introduction section, please single out only the main objectives of the study. The last two sentences look like conclusion , which can be confusing for readers. Please, revise this part. Other parts of Introduction are satisfactory, containing the most important findings regarding Acmella radicans regarding its components and biological activities. Authors have mentioned antioxidant, antibacterial and antiiflammatory action of this plant. Please, add some references regarding antiiflammatory action, including in vivo results.
Response 1. The paragraph has been corrected, retaining only the study objective to avoid confusion, Lines 112-114. In addition, studies related to anti-inflammatory action were added as suggested, Lines 76-98.
Comments 2. It is recommended to perform additional experiments regarding antioxidant potential, besides ABTS and DPPH. Eg. FRAP, inhibition of lipid peroxidation, O2-, H2O2, MDA….etc
Response 2. Thank you for the recommendation. We agree that there are many methods for evaluating antioxidant activity. However, we believe that the ABTS and DPPH methods are sufficient to support the antioxidant activity of the present study since it is not focused exclusively on antioxidant activity, but also on anti-inflammatory, antibacterial activity and chemical profile. Furthermore, the most widely applied and accepted methods are ABTS and DPPH, as these methods have the advantage of excellent stability under certain conditions, and both methods are complementary since ABTS can measure the activity of hydrophilic and lipophilic compounds, while DPPH can only dissolve in organic media. This has been widely reported in the literature and continues to be used in recent studies, for example: https://doi.org/10.1590/S0101-20612005000400016, https://doi.org/10.3303/CET25115004, https://doi.org/10.3390/ijms25168886 and https://doi.org/10.1016/j.fbio.2024.104706.
Comments 3. Please add some additional info about the normality of data distribution and tests used in the subsection: Statistical analysis
Response 3. Thank you for your comments. Information has been added regarding the normality tests for data distribution, which were performed using the Shapiro-Wilk method. This information was added on lines 633-638.
Comments 4. In the conclusion section, please mention the main finding regarding phytochemical components and antioxidant potential, you have focused only on antiiflammatory and antibacterial effects of radicans
Response 4. This was done, the conclusions were slightly modified. Lines 647 – 655.
Comments 5. The authors should discuss and compare the results of this study with others that not only investigated A.radicans but also its components alone. Try to explain which component is responsible for antioxidant, anti-inflammatory and antibacterial effect and the mechanisms of action as well
Response 5. Thank you for your comment. However, after careful review of the manuscript, we would like to clarify that the discussion does include possible compounds responsible for the observed antioxidant, anti-inflammatory, and antibacterial activities. Specifically, this discussion can be found in lines 305–307, 311–314, and 319–334 for anti-inflammatory activity; lines 355–360 for antibacterial activity; and lines 427–429 for antioxidant activity. In these sections, we mention that compounds such as alkamides, phenolic compounds, and certain triterpenes such as β-amyrin may be contributing to the biological effects observed.
Regarding the mechanism of action, we believe that a detailed discussion is premature in the context of this work, as the compounds identified by GC-MS have not yet been isolated or tested individually. To accurately determine their mechanisms of action, it is essential to obtain pure compounds and assess their biological activity separately. These studies are planned for future research, but they fall outside the scope of the present work.
Comments 6. Also, add future perspectives, and limitations of the study in the discussion section.
Response 6. This was done. A subsection of limitations and perspectives of the present study was added (2.7). Lines 434 - 450.
Comments 7. Please, add n- values in the captions of figures
Response 7. This was done, n-values (n = 3) were added in the figures and are highlighted in yellow in each figure caption where appropriate.
Comments 8. Comments on the Quality of English Language
Can be improved, please adress a native speaker or a professional to go throught the whole manuscript to correct minor mistakes, grammar and spelling.
Response 8. The manuscript was reviewed by a native English speaker. The certificate is attached.

Reviewer 2 Report
Comments and Suggestions for Authors
This study explores the phytochemical profile and biological activity of Acmella radicans, a traditionally used medicinal plant, by employing GC-MS analysis and in vitro assays. The researchers evaluated antioxidant, anti-inflammatory, and antibacterial properties of ethanolic extracts from aerial and root parts and their solvent-based fractions. Notably, the dichloromethane fractions exhibited potent inhibitory effects on NO and TNF-α production in LPS-stimulated RAW 264.7 macrophages. GC-MS identified high concentrations of spilanthol and other alkamides as likely bioactive compounds, especially in root extracts. Antibacterial assays further revealed promising activity against MRSA strains. These findings suggest A. radicans as a promising source of bioactive metabolites with potential therapeutic relevance, especially for inflammatory and resistant bacterial conditions.
However, before publication, the manuscript requires major revisions related to experimental clarity, consistency of data interpretation, deeper literature integration, improved figure presentation, and correction of multiple grammatical issues.
Comments for authors
Comment 1. The title is accurate in reflecting the study's aims. However, consider including “GC-MS-identified alkamides” to better highlight the chemical novelty and draw broader interest from pharmacognosy audiences..
Comment 2. While the introduction offers useful background, it lacks incorporation of more recent studies evaluating Acmella species in anti-inflammatory applications. Recent evidence suggests novel insights into alkamide signaling in inflammation—an area the current study could better contextualize within. A strategic update to this section would notably strengthen the paper.
Comment 3. The introduction lacks the recent literature on biomedical applications of plant-derived compounds. I recommend the inclusion of a very recent study in the introduction section:
J.N. Rana, S. Mumtaz, Prunin: An Emerging Anticancer Flavonoid, Int. J. Mol. Sci. 26 (2025). https://doi.org/10.3390/ijms26062678.
Comment 4. The bar plots in Figures 2–5 lack direct statistical comparisons between different fractions. Are there significant differences between AE, RE, DFAE, and DFRE at equivalent concentrations?
Comment 5. The link between identified alkamides and anti-inflammatory action is speculative. Provide references or propose possible receptor targets (e.g., cannabinoid or TRP channels) to justify mechanistic claims?
Comment 6. Why do DFAE and DFRE fractions cause >85% cytotoxicity at 100 μg/mL while showing anti-inflammatory benefits at 25 μg/mL?
Comment 7. The reported 50% lower antioxidant capacity of roots contrasts with the referenced IC₅₀ values, suggesting otherwise. Reconcile this inconsistency—was it due to differences in the extraction protocol?
Comment 8. The reported 50% lower antioxidant capacity of roots contrasts with the referenced IC50 values, suggesting otherwise. Reconcile this inconsistency—was it due to differences in the extraction protocol?
Comment 9. The conclusion suggests immediate therapeutic relevance. Consider softening this and emphasizing that in vivo validation and toxicity profiling are still pending.
Comment 10. Several grammatical errors, nonscientific phrasings (e.g., “a ributed,” “showed markedly suppression,” or “di erent fractions”) appear across the manuscript. A thorough professional proofreading or editing is recommended before publication.
End of the report!
Author Response
Response to Reviewer 2 Comments
This study explores the phytochemical profile and biological activity of Acmella radicans, a traditionally used medicinal plant, by employing GC-MS analysis and in vitro assays. The researchers evaluated antioxidant, anti-inflammatory, and antibacterial properties of ethanolic extracts from aerial and root parts and their solvent-based fractions. Notably, the dichloromethane fractions exhibited potent inhibitory effects on NO and TNF-α production in LPS-stimulated RAW 264.7 macrophages. GC-MS identified high concentrations of spilanthol and other alkamides as likely bioactive compounds, especially in root extracts. Antibacterial assays further revealed promising activity against MRSA strains. These findings suggest A. radicans as a promising source of bioactive metabolites with potential therapeutic relevance, especially for inflammatory and resistant bacterial conditions.
However, before publication, the manuscript requires major revisions related to experimental clarity, consistency of data interpretation, deeper literature integration, improved figure presentation, and correction of multiple grammatical issues.
Comments for authors
Comment 1. The title is accurate in reflecting the study's aims. However, consider including “GC-MS-identified alkamides” to better highlight the chemical novelty and draw broader interest from pharmacognosy audiences.
Response 1. This was done. We agree, the title was modified as suggested, adding “GC-MS-identified alkamides”.
Comment 2. While the introduction offers useful background, it lacks incorporation of more recent studies evaluating Acmella species in anti-inflammatory applications. Recent evidence suggests novel insights into alkamide signaling in inflammation—an area the current study could better contextualize within. A strategic update to this section would notably strengthen the paper.
Response 2. This was done. More recent studies were incorporated into the introduction section. Lines 77 - 99.
Comment 3. The introduction lacks the recent literature on biomedical applications of plant-derived compounds. I recommend the inclusion of a very recent study in the introduction section:
J.N. Rana, S. Mumtaz, Prunin: An Emerging Anticancer Flavonoid, Int. J. Mol. Sci. 26 (2025). https://doi.org/10.3390/ijms26062678.
Response 3. Thank you for the recommendation. However, we carefully reviewed the article and believe it is not relevant to include it in the present study, since our work did not address aspects of anticancer activity. However, we will consider it in our future related studies.
Comment 4. The bar plots in Figures 2–5 lack direct statistical comparisons between different fractions. Are there significant differences between AE, RE, DFAE, and DFRE at equivalent concentrations?
Response 4. Thank you for your comment. The original statistical analysis was designed to compare the effects of the ethanolic extracts and their respective fractions against the control group (untreated cells stimulated only with LPS) in terms of cell viability and NO production inhibition, as reported in the manuscript. However, as requested, we conducted an additional analysis using one-way ANOVA followed by Tukey’s multiple comparisons test to assess statistically significant differences between extracts and fractions of the same polarity at equivalent concentrations.
Regarding cell viability, significant differences were observed between ethanolic extracts at concentrations of 50 and 100 μg/mL; however, no significant differences were found among the corresponding fractions. In the case of NO production inhibition, significant differences were detected between extracts at 12.5, 25, and 50 μg/mL, but again, not among the fractions.
These additional results do not impact the interpretation or conclusions discussed in the manuscript; therefore, we consider it unnecessary to include them in the final version.
Comment 5. The link between identified alkamides and anti-inflammatory action is speculative. Provide references or propose possible receptor targets (e.g., cannabinoid or TRP channels) to justify mechanistic claims?
Response 5. Thank you for your comment. However, after careful review of the manuscript, we would like to clarify that the discussion does include possible compounds responsible for the observed anti-inflammatory activity. Specifically, this discussion can be found in lines 306 – 308, 312 – 315, and 316–335. Moreover, information has been added to the introduction section regarding the possible mechanisms specifically of alkamides such as spilanthol. Lines 76-98.
Regarding the mechanism of action of receptors as cannabinoids or propose possible receptor targets, we believe that a detailed discussion is premature in the context of this work, as the compounds identified by GC-MS have not yet been isolated or tested individually. To accurately determine their mechanisms of action, it is essential to obtain pure compounds and assess their biological activity separately. These studies are planned for future research, but they are outside the scope of this work as our objective was not to evaluate the mechanisms of action.
Comment 6. Why do DFAE and DFRE fractions cause >85% cytotoxicity at 100 μg/mL while showing anti-inflammatory benefits at 25 μg/mL?
Response 6. Thank you for your observation. Indeed, the results are correct, as it is well established that many molecules exhibit concentration-dependent cytotoxic effects; however, they can exert safe and effective therapeutic effects below their cytotoxicity threshold. In the case of the DFAE and DFRE fractions, at a concentration of 25 µg/mL, no cytotoxic effects were observed, and a significant inhibition of pro-inflammatory markers (nitric oxide and TNF-α) was achieved in RAW 264.7 cells, as indicated in the manuscript (Lines 192–196, 234–237 and 243 - 246). At 50 µg/mL, cytotoxicity remained minimal, with cell viability of 84.5 ± 8.3 % and 78.7 ± 3.0 % for DFAE and DFRE, respectively, suggesting an adequate safety margin for potential anti-inflammatory application These findings reinforce the evidence that the inhibition of NO production observed at 25 µg/mL is attributable to the extract’s effect and not to cytotoxic events such as cell death
Comment 7. The reported 50% lower antioxidant capacity of roots contrasts with the referenced IC₅₀ values, suggesting otherwise. Reconcile this inconsistency—was it due to differences in the extraction protocol?
Response 7. Thank you for your comment. As noted, the plant samples evaluated in this study were different from those used in the referenced work, as they were collected from distinct geographical locations. Therefore, variations in the results are expected. In addition, there were methodological differences: the referenced study employed ethanolic extraction through maceration at controlled temperature followed by sonication, whereas in the current study, only ethanolic maceration under controlled temperature was performed. Furthermore, ecological and edaphological differences between the collection sites may have contributed to the observed variations, as discussed in the manuscript. Lines 416 - 419.
Comment 8. The reported 50% lower antioxidant capacity of roots contrasts with the referenced IC50 values, suggesting otherwise. Reconcile this inconsistency—was it due to differences in the extraction protocol?
Response 8. This point has already been addressed in Response 7.
Comment 9. The conclusion suggests immediate therapeutic relevance. Consider softening this and emphasizing that in vivo validation and toxicity profiling are still pending.
Response 9. The conclusion section was modified as suggested. Lines 647 – 655.
Comment 10. Several grammatical errors, nonscientific phrasings (e.g., “a ributed,” “showed markedly suppression,” or “di erent fractions”) appear across the manuscript. A thorough professional proofreading or editing is recommended before publication.
Response 10. The manuscript was reviewed by a native English speaker. The certificate is attached.
End of the report!

Round 2
Reviewer 1 Report
Comments and Suggestions for Authors
The authors have adequately addressed all comments and concerns raised during the review process. The manuscript is now clear, scientifically sound, and meets the standards for publication. The authors have clearly invested significant effort into improving the manuscript, and I recommend acceptance in its current form.
Reviewer 2 Report
Comments and Suggestions for Authors
I recommend accepting the manuscript in its present form.